# The Relationship between Blood Lipids and Risk of Atrial Fibrillation: Univariable and Multivariable Mendelian Randomization Analysis

**DOI:** 10.3390/nu14010181

**Published:** 2021-12-31

**Authors:** Shengyi Yang, Rupak Pudasaini, Hong Zhi, Lina Wang

**Affiliations:** 1Key Laboratory of Environmental Medicine Engineering, Department of Epidemiology & Biostatistics, School of Public Health, Southeast University, Ministry of Education, Nanjing 210009, China; 220203857@seu.edu.cn (S.Y.); rupakaavas7@gmail.com (R.P.); 2Department of Cardiology, Zhongda Hospital, Southeast University, Nanjing 210009, China; 101005674@seu.edu.cn

**Keywords:** blood lipids, atrial fibrillation, Mendelian randomization, causal effect

## Abstract

We performed univariable and multivariable Mendelian randomization (MR) analysis to evaluate the association between blood lipids and risk of atrial fibrillation (AF), including low-density lipoprotein cholesterol (LDL-C), high-density lipoprotein cholesterol (HDL-C), triglyceride (TG), Apolipoprotein A1, and Apolipoprotein B. Methods: Data on the single nucleotide polymorphisms (SNPs) related to blood lipids were obtained from the UK Biobank study with more than 300,000 subjects of White British European ancestry, and data for AF were from the latest meta-analysis of Genome-wide association study (GWASs) with six independent cohorts with more than 1,000,000 subjects of European ancestry. The univariable MR analysis was conducted to explore whether genetic evidence of individual lipid-related traits was significantly associated with AF risks and multivariable MR analysis with three models was performed to assess the independent effects of lipid-related traits. Results: The IVW estimate showed that genetically predicted LDL-C (OR: 1.016, 95% CI: 0.962–1.073, *p* = 0.560), HDL-C (OR: 0.951, 95% CI: 0.895–1.010, *p* = 0.102), TG (OR: 0.961, 95% CI: 0.889–1.038, *p* = 0.313), Apolipoprotein A1 (OR: 0.978, 95% CI: 0.933–1.025, *p* = 0.356), and Apolipoprotein B (OR: 1.008, 95% CI: 0.959–1.070, *p* = 0.794) were not causally associated with the risk of AF. Sample mode (OR: 0.852, 95% CI: 0.731–0.993, *p* = 0.043) and weighted mode (OR: 0.907, 95% CI: 0.841–0.979, *p* = 0.013) showed that a 1-unit increase in TG (mmol/L) was causally associated with a 14.8% and 9.3% relative decrease in AF risk, respectively. The multivariable MR analysis with model 1, 2, and 3 indicated that TG, LDL-C, HDL-C, Apolipoprotein A1, and Apolipoprotein B were not associated with the lower risk for AF. Conclusions: Our multivariable Mendelian randomization analysis (MVMR) finding suggested no genetic evidence of lipid traits was significantly associated with AF risk. Furthermore, more work is warranted to confirm the potential association between lipid traits and AF risks.

## 1. Introduction

Atrial fibrillation (AF) is a common arrhythmia contributing to substantial social and medical burdens with significant health and socioeconomic impact [1]. The prevalence of AF is increasing, estimated to rise to 12.1 million in 2030 in the United States and 17.9 million in 2060 in the European Union [2]. AF is associated with high health system utilization, poor quality of life, and increased risk for hospitalization, heart failure, stroke, and death [3].

Significant effort has been made to define the underlying mechanisms of AF, such as fundamental electrophysiological and structural changes within the left atrium [4]. Notably, patients with high blood lipids levels can develop an inflammatory response in some cases [5], and hyperlipidemia appears to increase the risk of AF.

However, a few studies conducted to explore the relationship between blood lipids and AF have provided controversial results. In some observational studies, high levels of total cholesterol (TC) and low-density lipoprotein cholesterol (LDL-C) were unexpectedly identified to be inversely associated with the risk of AF [6,7,8,9,10]. However, the Multi-Ethnic Study of Atherosclerosis and the Framingham Heart Study found high-density lipoprotein cholesterol (HDL-C) and triglycerides (TG) were associated with the risk of AF except for LDL-C or TC [11]. Concurrently, low-level Apolipoprotein A1 and B were associated with increased risk of AF [12]. A systematic review of prospective studies found that serum TC, LDL-C, and HDL-C levels negatively correlated with risk of atrial fibrillation, while no significant correlation was found between TG levels and the incidence of AF [13]. Another meta-analysis reported a nonlinear association between TC and AF and a nonlinear association between LDL-C and the AF risks [14]. Notably, these studies included limited sample sizes with potential confounders.

Confirmation of a causal association is a challenging as the reverse causation and confounding between blood lipids and the risk of AF. Mendelian randomization (MR) has emerged as a powerful methodology for identifying the causation between exposures and diseases using genetic variants as instrument variables (IVs) [15]. MR analysis can largely overcome the confounders of individuals being randomly assigned genetic variants at the time of conception. Furthermore, the risk of reverse causation is also minimized, as the presence of a disease does not impact individuals’ genotypes [16].

In this study, we performed a univariable MR to explore whether genetic evidence of the lipid-related traits in individuals was significantly associated with AF risks. Meanwhile, each lipid-related entity can be expected to have its own influence or causal characteristics [17]. Therefore, we further preformed multivariable MR analysis with three models to evaluate the independent influence of lipid-related traits on AF using UKB data.

## 2. Materials and Methods

### 2.1. Data Resources and Study Design

Summary statistic data for LDL-C (N = 318,340) and HDL-C (N = 291,830), TG (N = 318,674), and Apolipoprotein A1 (N = 290,198) and Apolipoprotein B (N = 317,412) were from a meta-analyzed GWAS for 35 lab biomarkers from the UK Biobank (UKB) of White British European ancestry [18]. UK Biobank is a prospective cohort of over 500,000 men and women recruited and their health is being followed on a long term [19]. Data for AF were obtained from the latest meta-analysis of GWASs for AF with six independent cohorts (The Nord-Trøndelag Health Study, Michigan Genomics Initiative, DECODE, UK Biobank, DiscovEHR Collaboration Cohort, and AF Gen Consortium) with more than 1,000,000 subjects of European ancestry, including 60,620 cases with AF and 970,216 controls [20]. The details are presented in Table 1.

The univariable MR analysis were conducted to explore whether genetic evidence of individual lipid-related characteristics was significantly associated with AF risks and multivariable MR analysis with three models were conducted to assess the independent influence of lipid-related traits. In the model 1 of multivariable MR, pleiotropic effects across the included lipid traits were adjusted, including Apolipoprotein B, LDL-C, and TG for the causal associations with AF. In the model 2 of multivariable MR, pleiotropic effects across the included lipid traits including Apolipoprotein A1 and HDL-C were adjusted. Finally, in the model 3, we used the previously reported GWAS dataset for all circulating lipids traits [17].

### 2.2. Selection of Genetic Instrumental Variables

All genetic variants significantly associated with LDL-C and HDL-C, TG, and Apolipoprotein A1 and Apolipoprotein B levels (*p* < 5 × 10^−8^) were selected as IVs. The corresponding linkage disequilibrium was identified, we confirmed that the SNP was in a state of linkage disequilibrium, and the independence of the SNP was realized by cutting the SNP into a 10,000 kb window (*r*^2^ < 0.001) [21]. Then, the SNPs were removed that were related with potential confounders of the outcomes. In this study, blood pressure, blood sugar, BMI, chronic nephropathy, CAD, and CRP were identified as confounders (http://www.phenoscanner.medschl.cam.ac.uk/ accessed on 9 October 2021) [22]. SNP harmonization was performed to rectify the orientation of the alleles [15]. The details of IVs for LDL-C, HDL-C, TG, Apolipoprotein A1, and Apolipoprotein B in univariable MR analysis were presented in Appendix A. The SNPs of all lipid traits used in the multivariable MR analysis were acquired by clumping to a linkage disequilibrium threshold of *r*^2^ < 0.001. Finally, 301 SNPs were involved in model 1 of the multivariable MR analysis, 173 SNPs in model 2, and 437 SNPs in model 3. *F* statistic value for each instrument-exposure association was ranged from 28.422 to 49.559, demonstrating the smaller possibility of weak instrumental variable bias in the final results (Table 1).

### 2.3. Statistical Analysis

The way to obtain an MR estimate was to conduct an inverse variance weighted (IVW) meta-analysis of each Wald ratio [23]. When there was no evidence of targeted pleiotropy in the selected IVs (*p* for MR-Egger intercept > 0.05), the IVW approach was considered the most credible [24].

The weighted median method [25], sample mode method [26], weighted mode method [26], and MR-Egger method [25] were also utilized to evaluate the robust effects. The weighted median analysis can produce consistent estimates with at least 50% of the weight in the analysis coming from valid instrumental variables [27]. Cochran’s Q test was applied to assess heterogeneity of estimates of individual genetic variability. If the *p* value in the Cochran’s Q test was less than 0.05, the IVW with a multiplicative random-effects model in the eventual results was used; otherwise, a fixed-effects model was used [28]. The MR-Egger test was conducted to find out whether the main assumptions of MR were violated due to directional pleiotropy [25]. In MR-Egger test, the intercept evaluated the average pleiotropic effect of the genetic variation, and a value greater or less than zero indicated that the IVW estimate may be biased [29]. We also inspected potential directional pleiotropy based on the asymmetry of the funnel plots. Finally, MR-PRESSO was performed to validate the results in the IVW model, which detected and corrected the effects of outliers, generating reliable causal estimates of heterogeneity [30].

### 2.4. Sensitivity Analysis

The leave-one-out sensitivity analyses were conducted to evaluate the stability of results. *R*-squared was calculated to estimate the proportion of variance in outcomes, and the *F*-statistic value was calculated to mitigate the bias and predict the intensity of IVs.

Given the genetic and phenotypic relevance of lipid properties as a prior study reported [17], we further used multivariable IVW method with three models to disentangle the effects of different lipid-traits on AF. Furthermore, we performed a linear regression-based approach to estimate each risk factor separately [31].

All analyses were performed using the package “Two-Sample-MR” (version 0.5.6, Bristol, UK) and “MR-PRESSO” (version 1.0, New York, NY, USA) in R (version 4.0.5, Vienna, Austria).

## 3. Results

### 3.1. Univariable MR Analysis of Lipid Traits on AF Risks

Figure 1 reports the univariable MR estimated of lipid traits on AF risks. The IVW estimate showed that genetically predicted LDL-C (OR: 1.016, 95% CI: 0.962–1.073, *p* = 0.560), HDL-C (OR: 0.951, 95% CI: 0.895–1.010, *p* = 0.102), TG (OR: 0.961, 95% CI: 0.889–1.038, *p* = 0.313), Apolipoprotein A1 (OR: 0.978, 95% CI: 0.933–1.025, *p* = 0.356), and Apolipoprotein B (OR: 1.008, 95% CI: 0.959–1.070, *p* = 0.794) were not significantly associated with the risk of AF (Figure 1). The results were consistent in weighted median methods and weighted mode methods (Figure 1). However, sample mode (OR: 0.852, 95% CI: 0.731–0.993, *p* = 0.043) and weighted mode (OR: 0.907, 95% CI: 0.841–0.979, *p* = 0.013) showed that a 1-unit increase in TG (mmol/L) was causally associated with a 14.8% and 9.3% relative decrease in AF risk, respectively. Furthermore, the MR-PRESSO process verified the negative results (Table 2).

There were potential heterogeneities but no directional pleiotropies for the analysis results (Appendix A). The scatter plots and forest plots were displayed in Appendix A. The funnel plots were symmetrical (Appendix A) and the leave-one-out method indicated that no SNP was substantially driving the association between lipids traits and AF risks (Appendix A).

### 3.2. Multivariable MR Analysis in Model 1

In model 1 with mutual adjustment for LDL-C, TG, and Apolipoprotein B, the association between LDL-C and risk of AF was still non-significant (N = 301, OR = 0.972, 95% CI: 0.505–1.439, *p* = 0.891). It also showed that genetically predicted TG and Apolipoprotein B were not significantly associated with risk of AF (Figure 2, Appendix A). This negative effect was also found in a linear regression-based approach (Appendix A).

### 3.3. Multivariable MR Analysis in Model 2

In model 2 with mutual adjustment for HDL-C and Apolipoprotein A1, the association between Apolipoprotein A1 and risk of AF was still non-significant (N = 173, OR = 1.006, 95% CI: 0.867–1.145, *p* = 0.950, Figure 2, Appendix A). The result was consistent in complementary analyses using a linear regression-based approach (Appendix A).

### 3.4. Multivariable MR Analysis in Model 3

In model 3 with mutual adjustment for LDL-C, TG, and Apolipoprotein A1, HDL-C, and Apolipoprotein A1, there was no association between lipid with the risk of AF (Figure 2, Appendix A). The result was consistent in complementary analyses using a linear regression-based approach (Appendix A).

## 4. Discussion

Using an integrated approach, including conventional multivariate MR, our study aimed to test for a cause effect between genetically determined lipid traits and AF risks. However, our study showed there was no cause effect between them.

Observational studies have provided discrepant results on the relationship between LDL-C, HDL-C, TG, Apolipoprotein A1, and Apolipoprotein B and AF risk. For example, in a pool analysis of two community-based cohorts, HDL-C and TG were associated with the risk of AF but not LDL-C or TC [11]. In the Niigata Preventive Medicine Study, high levels of TC, LDL-C, and HDL-C were found to be associated with an increased risk of AF among Japanese [6], while, in addition to HDL-C and TG, higher levels of TC and LDL-C were found related with the increased risk of AF among Chinese [32]. Lacking adjustments including obesity, geographical and ethnic variations, and other CVD risk factors may partly explain inconsistencies among studies. A meta-analysis of large cohort studies reported that serum TC, LDL-C, and HDL-C levels negatively correlated with risk of AF, while no significant correlation was found between TG levels and incident AF [13].

It seems there was a “cholesterol paradox” in AF [10]. The potential mechanisms of the identified inversed causal effects of LDL-C and TC were as follows. Firstly, it is an alteration of cardiac ion channels caused by cholesterol [33,34,35,36]. Some studies have suggested that cholesterol regulates the distribution and function of the Kv1.5 K^+^, Kir2.1 K^+^, and Na^+^ channel, and participates in the etiopathogenesis of AF. Secondly, it might be confounded by hyperthyroidism status, which was found to reduce LDL-C levels and associated with risk of AF [37,38,39]. Another confounding factor might be the natriuretic peptides (NT-proBNP or BNP). It has been described that there is a negative correlation between LDL-C levels and NT-proBNP [40], and natriuretic peptides are powerful predictors of AF risk [11,41,42,43]. Thirdly, patients with elevated TC levels have an anti-inflammatory effect in certain situations, which might also be related in the etiopathogenesis of AF [44]. With the univariable MR analysis, we have proven that genetically predicted TG was associated with the lower risk for AF. This might be explained by the confounder factors including other lipid traits. Furthermore, multivariable MR analysis indicated that there was no correlation between lipid and AF risks and the potential mechanisms of lipid and AF are not yet fully elucidated.

Both LDL-C and TG are transported in atherogenic lipoproteins, each of which contains Apolipoprotein B molecule [45,46]. Some cross-sectional studies [47,48] and prospective observational studies [49] suggested Apolipoprotein B might be a more accurate cardiovascular risk marker than total cholesterol or LDL-C levels. The MR studies have added evidence that the quantity of Apolipoprotein B particles within the arterial lumen is the most immediate indicator of atherosclerotic damage, and that Apo B particles cause damage to the arterial wall. Given correlation among lipid-related characteristics, multivariable MR analysis with three models, were designed to test the independent effects of each lipid trait. However, our results indicated that high Apolipoprotein B levels were not associated with the increased risk of AF both in univariable and multivariable MR analysis. Increased TG levels have been shown in epidemiological and clinical studies to be a biomarker of cardiovascular (CV) risk [50], but we found that there was no causal relationship between TG and AF risks.

Our MVMR analyses provided the genetic evidence that none of the lipid traits was significantly associated with AF risks. There are some strengths in our study. Firstly, data were obtained from different samples, genetic associations can be gained from large GWAS, which considerably improves the statistical power for the detection of small effects in complex phenotypes [51]. Secondly, the genetic variants were distributed on separate chromosomes, and underlying gene-gene interaction might have little influence on the effects [52].

There are several limitations to our study. Firstly, there was heterogeneity among our results. Due to the GWAS data, any potential nonlinear relationships or stratification effects which differs by health status, age, or gender cannot be examined. This may be the resource of heterogeneity. Secondly, despite the lack of targeted pleiotropy indication in the analysis, we could not exclude the association, which is almost completely mediated through other causal pathways. Thirdly, we did not explore the association between blood lipids and different AF subtypes. Finally, our datasets included mostly European populations which limited applicability of results for non-European populations.

## 5. Conclusions

In conclusion, our MVMR analyses provided genetic evidence that no genetically determined lipid traits were significantly associated with AF risks. more work is warranted to confirm the potential association between lipid traits and AF risks.

## Figures and Tables

**Figure 1 nutrients-14-00181-f001:**
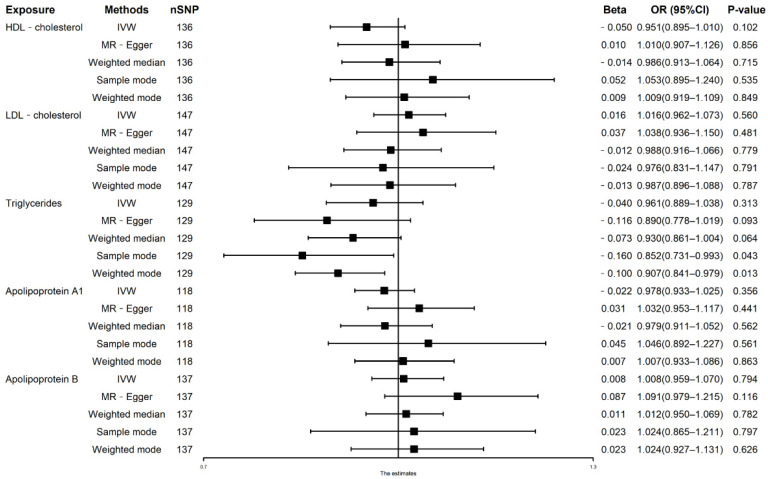
Associations of HDL-C, LDL-C, triglycerides, Apolipoprotein A1, and Apolipoprotein B with AF in univariable Mendelian randomization analysis. HDL, high-density lipoprotein; LDL, low-density lipoprotein; IVW, inverse variance weighted.

**Figure 2 nutrients-14-00181-f002:**
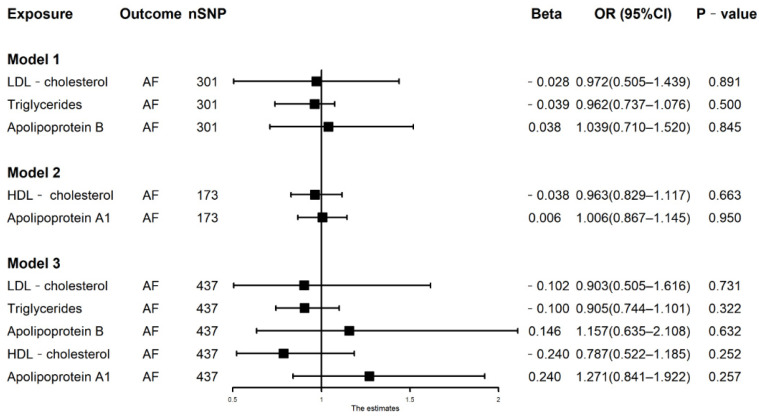
Model 1, associations of LDL-C, TG, Apolipoprotein B with AF risks in multivariable MR analysis. Model 2, associations of HDL-C, Apolipoprotein A1 with AF risks in multivariable MR analysis. Model 3, associations of HDL-C, LDL -C, TG, Apolipoprotein A1, and Apolipoprotein B with AF in multivariable MR analysis. HDL-C, high-density lipoprotein cholesterol; LDL, low-density lipoprotein cholesterol.

**Table 1 nutrients-14-00181-t001:** Details of studies included and predictive strength of IVs in Mendelian randomization analyses (two-sided α = 0.05).

Exposures/Outcomes	Consortium	Ethnicity	Sample Sizes	R-Squared % (of Variance in AF)	F-Statistic (Total)
HDL-C	UKB	European	291,830	1.582	34.375
LDL-C	UKB	European	318,340	1.296	28.422
TG	UKB	European	318,674	1.543	38.704
Apolipoprotein A1	UKB	European	290,198	1.976	49.559
Apolipoprotein B	UKB	European	317,412	1.354	31.788
Atrial fibrillation	HUNT, DECODE, DiscovEHR, MGI, UKB, and AF Gen Consortium	European	1,030,836	NA	NA

AF, atrial fibrillation; UKB, UK Biobank; HDL-C, high density lipoprotein cholesterol; LDL-C, low density lipoprotein cholesterol; TG, Triglycerides; HUNT, The Nord-Trøndelag Health Study; DECODE, DiscovEHR, Collaborative analysis of Diagnostic criteria in Europe study; MGI, Michigan Genomics Initiative; AF Gen, Atrial Fibrillation Genetics.

**Table 2 nutrients-14-00181-t002:** MR-PRESSO for causal effect between circulation bilirubin levels and AF.

Exposure	Raw Estimates				Outlier Corrected Estimates				Distortion Test
	nSNP	Beta	OR (95%CI)	*p*-Value	nSNP	Beta	OR (95%CI)	*p*-Value	*p*-Value
HDL-C	138	−0.025	0.975(0.920,1.030)	0.379	137	−0.018	0.982(0.928,1.036)	0.517	0.695
LDL-C	147	−0.001	0.999(0.947,1.051)	0.966	145	−0.004	0.996(0.946,1.046)	0.882	0.942
TG	129	−0.041	0.960(0.885,1.035)	0.291	125	−0.047	0.954(0.897,1.011)	0.114	0.847
Apolipoprotein A1	118	−0.002	0.998(0.994,1.002)	0.921	NA	NA	NA	NA	NA
Apolipoprotein B	137	0.008	1.008(0.959,1.070)	0.786	NA	NA	NA	NA	NA

HDL-C, high density lipoprotein cholesterol; LDL-C, low density lipoprotein cholesterol; TG, Triglycerides; SNP, single nucleotide polymorphisms; OR, odds ratio.

## Data Availability

GWAS dataset are available at ieu open GWAS project (https://gwas.mrcieu.ac.uk/, last accessed 10 September 2021).

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
