# Peer review of "The Relationship between Blood Lipids and Risk of Atrial Fibrillation: Univariable and Multivariable Mendelian Randomization Analysis"

_nutrients, 2021, doi:10.3390/nu14010181_

Round 1

Reviewer 1 Report

The authors present data that demonstrates that genetic-mediated blood lipids are not linked to atrial fibrillation risk. While the data seem to support the overall conclusions, I would recommend more discussion and perhaps introduction into the topic of GWAS. 

  1. The authors mention their GWAS data but only talk briefly about this in their manuscript. I had to go to their supplemental tables to learn more and would have liked more intro and discussion as to how GWAS was integrated into their statistical design and how their data demonstrate that genetics is not a determinate of AF linked to blood lipids.
  2. It's unclear how the UKB data and the AF data from the other datasets were linked. Did the other datasets look only at AF or did they also look at blood lipids? If they didn't look at blood lipids how can you perform correlation analysis? This was unclear in the methods.

Author Response

Reviewer 1

Comment 1: The authors mention their GWAS data but only talk briefly about this in their manuscript. I had to go to their supplemental tables to learn more and would have liked more intro and discussion as to how GWAS was integrated into their statistical design and how their data demonstrate that genetics is not a determinate of AF linked to blood lipids.

Response: Thanks for your suggestion. We have added the detailed information on the data selection in the section of Methods:

“All the summary data for LDL-C and HDL-C, TG, and Apolipoprotein A1 and Apolipoprotein B was selected from the UKB GWAS datasets we mentioned in the section of Methods. (lines 70-72, page 2) Firstly, the SNPs associated with lipids at genome-wide significance (p < 5 × 10 -8) in the GWAS datasets were selected as IVs. And the corresponding linkage disequilibrium was tested. Then, the SNPs were excluded that were associated with potential confounders of the outcomes. Next, we extracted the SNP-related imformation from AF GWAS dataset. Finaly, we ingtegrated the SNP-exposure effects and the SNP-outcome effects for MR analysis (Supplementary table 1-5), and the SNP harmonization was conducted to correct the orientation of the alleles.

In order to demonstrate that genetics is not a determinate of AF linked to blood lipids, we did several MR analysis methods, including IVW meta-analysis of each Wald Ratio, The weighted median method, sample mode method, weighted mode method and MR-egger method. Finally, MR-PRESSO was performed to validate the results in IVW model, which detected and corrected the effects from outliers, yielding causal estimates that are robust to heterogeneity. After univariable MR analysis, multivariable MR analysis with three models were used to evaluate the independent effects of lipid-related traits. Finally, we demonstrated that no genetic evidence of lipid traits was significantly associated with AF risk.

Comment 2: It's unclear how the UKB data and the AF data from the other datasets were linked. Did the other datasets look only at AF or did they also look at blood lipids? If they didn't look at blood lipids how can you perform correlation analysis? This was unclear in the methods.

Response: Thanks for your question. This was a two-sample MR analysis, and the SNP-exposure effects and the SNP-outcome effects were obtained from the two original studies. Summary statistic data of lipid was from UK Biobank which was a prospective cohort that recruited more than 500,000 men and women and their health is being followed on a long term. And Data for AF was obtained from the latest meta-analysis of GWASs for AF with six independent cohorts with more than 1,000,000 subjects of European ancestry, including 60,620 cases with AF and 970,216 controls (lines 70-73, page 2). Currently speaking, these two studies have the largest sample sizes and the robust associations on the genetic variants and lipid or AF risks.which have the tremendous advantage that causal inference can be made between two traits even if they aren’t measured in the same set of samples, enabling us to harness the statistical power of pre-existing large GWAS analyses (1-2).

(1) Hemani G, Zheng J, Elsworth B, et,al. The MR-Base platform supports systematic causal inference across the human phenome. Elife. 2018 May 30;7:e34408. doi: 10.7554/eLife.34408. PMID: 29846171; PMCID: PMC5976434.

(2) Pierce BL, Burgess S. Efficient design for Mendelian randomization studies: subsample and 2-sample instrumental variable estimators. Am J Epidemiol. 2013 Oct 1;178(7):1177-84. doi: 10.1093/aje/kwt084. Epub 2013 Jul 17. PMID: 23863760; PMCID: PMC3783091.

So we did not choose the other datasets to conduct the MR causal inference on the lipid and AF risks.

Reviewer 2 Report

Yang and colleagues conducted a Mendelian randomization evaluation of genetically-controlled blood lipid traits in association with atrial fibrillation. No causal association of the lipid or lipoprotein parameters was found. The methods generally are appropriate and the analyses used large patient datasets from multiple international studies of Caucasian subjects. A few improvements would aid the paper:

Major Comments:

  1. It is not clear in the abstract what the sample sizes are or what the source populations are in the study. Not all details from Table 1 are needed in the abstract, but something more would be appropriate to inform the reader and attract their attention to the paper.
  2. In the abstract Results section and in the main text Results section 3.1 the odds ratios are presented without any context of what units are being described. Are the ORs per mg/dL or mmol/L, or per 10 units, or such? It isn't clear how to interpret the ORs.
  3. In the Methods, section 2.1,  the numbers in parentheses in the first sentence are not labeled. After looking at them for a minute they appear to be sample sizes, but could plausibly be means or medians of something, so labeling them is important for clarity.
  4. In section 2.2, lines 100 and 107, the r2 values for pruning are listed as r2<0.001, but it seems that usually the SNPs would be pruned if they have greater r2. One convention is r2>0.001, which is the opposite of what is written. Please check this.

Author Response

Reviewer 2

Comment 1: It is not clear in the abstract what the sample sizes are or what the source populations are in the study. Not all details from Table 1 are needed in the abstract, but something more would be appropriate to inform the reader and attract their attention to the paper.

Response: Thanks for your suggestions. We have added more details of the papulation we included in the abstract as following:

“Data on the single nucleotide polymorphisms (SNPs) related to blood lipids was obtained from UK Biobank study with more than 300,000 subjects of White British European ancestry, and data for AF was from the latest meta-analysis of GWASs with six independent cohorts with more than 1,000,000 subjects of European ancestry, respectively.” (lines 14-17, page 1).

Comment 2: In the abstract Results section and in the main text Results section 3.1 the odds ratios are presented without any context of what units are being described. Are the ORs per mg/dL or mmol/L, or per 10 units, or such? It isn't clear how to interpret the ORs?

Response:Thanks, we have interpretedthe OR values in abstract and in the main text results section 3.1 as following:

“Sample mode (OR: 0.852, 95% CI: 0.731-0.993, P=0.043) and weighted mode (OR: 0.907, 95% CI: 0.841-0.979, P=0.013) showed that a 1 unit increase in TG (mmol/L) was caus-ally associated with a 14.8% and 9.3% relative decrease in AF risk, respectively.(lines 25-27, page 1 and lines 155-158, page 4).”

Comment 3: In the Methods, section 2.1,  the numbers in parentheses in the first sentence are not labeled. After looking at them for a minute they appear to be sample sizes, but could plausibly be means or medians of something, so labeling them is important for clarity.

Response: Thanks for your correction. The numbers had been labeled as the sample sizes for corresponding lipids traitin lines 77-78, page 2.

Comment 4: In section 2.2, lines 100 and 107, the r2 values for pruning are listed as r2<0.001, but it seems that usually the SNPs would be pruned if they have greater r2. One convention is r2>0.001, which is the opposite of what is written. Please check this.

Response: Thanks. We conducted the linkage disequilibrium analysis to confirm that there were any SNPs in a linkage disequilibrium state and defined the r2< 0.001 threshold to selected SNPs (1).

(1) Park S, Lee S, Kim Y. Atrial fibrillation and kidney function: a bidirectional Mendelian randomization study. Eur Heart J. 2021 Jul 31;42(29):2816-2823. doi: 10.1093/eurheartj/ehab291. PMID: 34023889.